# Selective Reagent Ion-Time-of-Flight-Mass Spectrometric Investigations of the Intravenous Anaesthetic Propofol and Its Major Metabolite 2,6-Diisopropyl-1,4-benzoquinone

Anesu Chawaguta [1,*] , Florentin Weiss [1] , Alessandro Marotto [2], Simone Jürschik [1] and Chris A. Mayhew [1]

1   Institute for Breath Research, University of Innsbruck, Innrain 66 and 80/82, A-6020 Innsbruck, Austria
2   Institute of Organic Chemistry, University of Innsbruck, Innrain 80/82, A-6020 Innsbruck, Austria
*   Correspondence: anesu.chawaguta@uibk.ac.at

**Featured Application: First detailed reduced electric field selected reagent ion mass spectrometric study of propofol and its major metabolite for potential analytical breath test applications in the clinical environment.**

**Abstract:** The first detailed selected reagent ion-time-of-flight-mass spectrometric fundamental investigations of 2,6-diisopropylphenol, more commonly known as propofol ($C_{12}H_{18}O$), and its metabolite 2,6-diisopropyl-1,4-benzoquinone ($C_{12}H_{16}O_2$) using the reagent ions $H_3O^+$, $H_3O^+.H_2O$, $O_2^{+\bullet}$ and $NO^+$ are reported. Protonated propofol is the dominant product ion resulting from the reaction of $H_3O^+$ with propofol up to a reduced electric field strength (*E/N*) of about 170 Td. After 170 Td, collision-induced dissociation leads to protonated 2-(1-methylethyl)-phenol ($C_9H_{13}O^+$), resulting from the elimination of $C_3H_6$ from protonated propofol. A sequential loss of $C_3H_6$ from $C_9H_{13}O^+$ also through collision-induced processes leads to protonated phenol ($C_6H_7O^+$), which becomes the dominant ionic species at *E/N* values exceeding 170 Td. $H_3O^+.H_2O$ does not react with propofol via a proton transfer process. This is in agreement with our calculated proton affinity of propofol being 770 kJ mol$^{-1}$. Both $O_2^{+\bullet}$ and $NO^+$ react with propofol via a charge transfer process leading to two product ions, $C_{12}H_{18}O^+$ (resulting from non-dissociative charge transfer) and $C_{11}H_{15}O^+$ that results from the elimination of one of the methyl groups from $C_{12}H_{18}O^+$. This dissociative pathway is more pronounced for $O_2^{+\bullet}$ than for $NO^+$ throughout the *E/N* range investigated (approximately 60–210 Td), which reflects the higher recombination energy of $O_2^{+\bullet}$ (12.07 eV) compared to that of $NO^+$ (9.3 eV), and hence the higher internal energy deposited into the singly charged propofol. Of the four reagent ions investigated, only $H_3O^+$ and $H_3O^+.H_2O$ react with 2,6-diisopropyl-1,4-benzoquinone, resulting in only the protonated parent at all *E/N* values investigated. The fundamental ion-molecule studies reported here provide underpinning information that is of use for the development of soft chemical ionisation mass spectrometric analytical techniques to monitor propofol and its major metabolite in the breath. The detection of propofol in breath has potential applications for determining propofol blood concentrations during surgery and for elucidating metabolic processes in real time.

**Keywords:** PTR-ToF-MS; SRI-ToF-MS; propofol; 2,6 diisopropyl-1,4-benzoquinone; intravenous anaesthetic; reduced electric field; $H_3O^+$; $NO^+$; $O_2^{+\bullet}$

## 1. Introduction

Propofol (2,6-diisopropylphenol, $C_{12}H_{18}O$) is a commonly used short-acting but potent intravenous hypnotic drug and is generally used to induce procedural sedation and maintain general anaesthesia. A key issue with an intravenous anaesthetic is that its plasma concentration cannot be easily and rapidly monitored during surgery. Although no direct assessment of the plasma levels to ensure adequate anaesthesia is possible, these levels could be indirectly determined in real time through an analysis of the concentrations of propofol contained in the exhaled breath of patients undergoing surgery.

Soft chemical ionisation spectrometric measurements of the intravenous general anaesthetic propofol in breath and within a clinical environment were initiated twenty years ago using proton transfer reaction mass spectrometry (PTR-MS) by Harrison et al. [1]. This 2003 study provided the "seminal investigation [that] gave proof of concept of reality to the conjecture of pulmonary propofol elimination and its measurement" [2]. A number of the follow-up soft chemical ionisation spectrometric studies used mass spectrometry, including PTR-MS, Ion-Molecule Reaction Mass Spectrometry (IMR-MS) and Selected Ion Flow Tube Mass Spectrometry (SIFT-MS), to provide the *m/z* values of the product ions [3–8]. Other soft chemical ionisation analytical techniques are based on the applications of ion mobility spectrometry (IMS), for which drift times (reduced ion mobilities) of the product ions are used to identify the presence of propofol. The first IMS exhaled propofol breath studies are those reported by Perl et al. [9,10]. These were followed by further work led by Baumbach [11–13], that ultimately led to the development of a clinical ion mobility spectrometric device called the Exhaled Drug MONitor (EDMON), supplied by B. Braun Melsungen AG, Melsungen, Germany. This instrument is based on multi-capillary column–ion mobility spectrometry (MCC-IMS), as described in recent publications by Braathen et al. [14] and Teucke et al. [15].

The considerable interest in the measurement of the intravenous anaesthetic propofol in exhaled breath relates to the potential of converting breath to plasma concentrations in order to monitor the anaesthetic dose during surgeries. With regards to this aspect, in his 2007 editorial review, Kharasch raised the important point of the need to monitor carbon dioxide simultaneously with propofol in exhaled breath in real-time in order to permit the end-tidal phase of the breath to be defined [2]. However, this is not possible to do with MCC-IMS techniques but is relatively easy to undertake with analytical techniques such as PTR-MS and $CO_2$ sensors. It is, therefore, surprising that there have been no detailed PTR-MS follow-up studies to the 2003 proof-of-principle report of propofol by Harrison et al. [1], especially given that only nominal *m/z* values of the product ions were provided at only one reduced electric field value (at approximately 140 Td, where 1 Td = $10^{-17}$ V cm$^2$). (The reduced electric field is the ratio of the electric field strength, *E*, to the molecular number density, *N*, in the drift tube).

Harrison et al. could only provide nominal *m/z* values because of the low resolving power of the quadrupole mass spectrometer provided in their PTR-MS. The main goal of this paper is to deliver for the first time a more in-depth PTR-MS study that provides not only highly accurate *m/z* values of the product ions but also information on the product ion distributions over a range of *E/N* values (approximately 60–210 Td). In addition to providing details on the product ions resulting from the reactions of $H_3O^+$ with propofol, we have significantly extended the original Harrison et al. study by using a proton transfer reaction-time-of-flight mass spectrometer (PTR-ToF-MS) in so-called selected reagent ion (SRI) mode, which is referred to as SRI-ToF-MS. This has allowed us to investigate the reactions of two other commonly used reagent ions in soft chemical ionisation mass spectrometry, namely $NO^+$ and $O_2^{+\bullet}$, and with their associated product ion distributions being also provided as a function of *E/N*.

Propofol is efficiently metabolised in the liver. Although the main metabolite of propofol, 2,6 diisopropyl-1,4-benzoquinone, is tentatively identified in the mass spectra of breath samples analysed by Harrison et al., there has been no independent study of the metabolite to confirm this. We have therefore included in this investigation a PTR/SRI-ToF-MS study of that compound for completeness.

This paper provides much-needed fundamental knowledge on the ion-molecule chemistry occurring in the drift tube of the SRI-ToF-MS for potential analytical chemistry applications to the real-time monitoring of propofol and its major metabolite in exhaled breath. This is important not only for its potential use to determine plasma concentrations continuously during surgery (as mentioned above), which could be used to provide feedback-controlled anaesthesia [16,17], but also for investigations of the metabolic processes resulting in the production of the metabolite and the elimination of propofol from the human body post-surgery. Post-surgery measurements could be used, for example, to provide information on abnormal metabolic stresses [6]. Given the potential breath analysis applications, we have

undertaken many of our measurements under what we have described as *normal* and *humid* drift tube conditions [18]. The *humid* drift tube conditions were applied in order to mimic the operational conditions that will occur for direct breath sampling, i.e., for which the water content in the exhaled breath has not been removed prior to the sample entering the drift tube. Such investigations provide much-needed information on potential analytical problems, such as whether secondary product ions occur that result from the reactions of the primary product ions with the $H_2O$ present in the drift tube. This can lead to a reduction in analytical sensitivity for the detection of a compound, as has been found to occur for the detection of a number of compounds [19,20], and in the context of this study relating to anaesthetics, a number of volatile inhalation anaesthetics, namely isoflurane, enflurane, sevoflurane and desflurane [18,21–23].

## 2. Materials and Methods

### 2.1. Propofol

Propofol (CAS: 2078-54-8, molar mass 178.275 g mol$^{-1}$)) with a stated purity of at least 97% was purchased from SAFC Supply Solutions (St. Louis, MO, USA), and it was used without any additional purification.

### 2.2. 2,6-Diisopropyl-1,4-benzoquinone

Given that 2,6-diisopropyl-1,4-benzoquinone is not readily available commercially, we decided to synthesise the compound. The method of synthesis is briefly described. Propofol (1.00 g; 5.6 mmol) was added to acetonitrile (40 mL) at room temperature under an argon atmosphere with ceric ammonium nitrate (6.14 g; 11.2 mmol) having been added. The reaction mixture was then heated to 80 °C for 1 h. The solvent was removed under reduced pressure. Water (20 mL) and dichloromethane (20 mL) were then added to the residue that was and then transferred into a separatory funnel for separation. The aqueous phase was then extracted with dichloromethane (3 × 20 mL). The combined organic layers were dried over anhydrous sodium sulfate, filtered and dried via evaporation. The crude residue was purified by column chromatography, using a gradient of ethyl acetate in hexane (starting from 5% and then gradually increased to 15% within a period of 20 min) to give 2,6-diisopropyl-1,4-benzoquinone as a viscous oil. The purity of the synthesised substance was determined to be >90%, with the major impurity being propofol, using NMR analysis details which are presented in Supplementary Figure S1.

### 2.3. Sample Preparation

Given their low vapour pressure, reliable direct headspace measurements of propofol and 2,6-diisopropyl-1,4-benzoquinone are difficult. Therefore, we adopted a technique more suitable for the chemical analysis of compounds that are semi-volatiles at room temperature. A 1 L glass gas bulb (Supelco Analytical, PA, USA) was placed in an oven (Memmert GML, Innsbruck, Austria) at 60 °C. This bulb was evacuated using a vacuum pump (Vacuumbrand GmbH + Co KG, Wertheim, Germany) for about 30 min. Following this, 1 µL of neat propofol or its metabolite was injected into the preheated and evacuated gas bulb through a septum. The gas bulb was then connected via a valve to a 3 L Tedlar gas bag (SKC Ltd., Dorset, UK) which was filled with high-purity nitrogen (99.9999%, <3 ppmv water). Once the valve was opened, the gas bulb was brought-up to atmospheric pressure. To limit condensation of the compounds, the gas bulb was maintained at 60 °C throughout the measurements.

In order to obtain a constant compound volume mixing ratio (~600 ppbv) in the drift tube, 1.25 mL of the standard from the gas bulb was introduced into a 250 mL glass syringe filled with either dry (~0% relative humidity (rH)) or humid (~100% rH) high-purity nitrogen for the normal and humid drift tube condition measurements, respectively. The humid sample, which mimics breath samples, was only used for the propofol measurements and was achieved by bubbling the high-purity nitrogen through distilled water kept at room temperature. The 250 mL syringe was also held at 60 °C throughout the measurements.

*2.4. The Proton Transfer Reaction-Time-of-Flight-Mass Spectrometer/Selective Reagent Ionisation-Time-of-Flight-Mass Spectrometer*

The PTR/SRI-ToF-MS used in this study was a compact high-performance extended PTR-TOF 6000 X2 (Ionicon Analytik GmbH, Innsbruck, Austria) [24–27] that has a high mass resolving power (m/$\Delta$m) of approximately 6000 at $m/z$ 147, which provides high confidence in accurately identifying the product ions from the $m/z$ values. Information on the operational procedures of PTR/SRI-ToF-MS is discussed in detail in the literature [28,29]. In brief, the drift tube was maintained at a constant pressure of 2.6 mbar for $H_3O^+$ and 2.3 mbar for $O_2^{+\bullet}$ and $NO^+$ using a dry or humid nitrogen carrier gas and at a temperature of 353 K. Owing to the fixed number density, $N$, only the drift tube voltage needed to be varied to alter the value of the reduced electric field. By varying the applied drift voltage from about 220 V up to about 660 V, an $E/N$ range of approximately 60–210 Td was covered.

The reagent ions $H_3O^+$ and $H_3O^+.H_2O$ (PTR-mode) and $NO^+$ and $O_2^{+\bullet}$ (SRI-mode) investigated in this study were created in the hollow cathode ion source by introducing water vapour, a 3:1 mix of nitrogen (99.9999% purity) and medical grade oxygen (>99.5% purity), into the ion source, respectively. Figure 1 displays the signal intensities in counts per second (cps) for all reagent ions as a function of $E/N$ ranging from about 60 Td up to about 210 Td in steps of 20 Td under both *normal* and *humid* drift tube conditions. In contrast to SIFT-MS, for which the reagent ion is actively selected [30], soft chemical ionisation mass spectrometric analytical instruments rely solely on ion chemistry to form the dominant reagent ion. As a consequence of this, impurity reagent ions are always present in the drift tube but generally in very low concentrations. The only exception is $H_3O^+.H_2O$, whose levels can be higher than those of $H_3O^+$ depending on the $E/N$ value applied and the humidity of the buffer gas within the drift tube. It is the responsibility of the experimentalist to ensure that the contributions of the impurity reagent ions to product ion formation are either negligible or properly accounted for. As evident from Figure 1, many of the impurity reagent ions contribute less than about 3% to the total reagent ion intensity at any reduced electric field value used in this study. They can therefore be safely ignored. The only exception to this is for the case of proton transfer studies when $H_3O^+$ is the preferred reagent ion. In our study, at low reduced electric field strengths, namely less than approximately 70 Td and 120 Td for normal and humid drift tube conditions, respectively, the protonated water dimer, $H_3O^+.H_2O$, becomes more dominant than $H_3O^+$ (compare Figure 1a with Figure 1b).

## 3. Results and Discussions

No significant changes in the product ion distributions were observed between the *humid* and the *normal* operating drift tube conditions. This implies firstly that when operating the instrument in PTR mode, $H_3O^+.H_2O$ does not react with propofol and secondly that there are no secondary reactions of neutral water in the drift tube with any of the product ions that result from the reactions of all three reagent ions with propofol. For this reason, only the product ion intensities and their dependence on $E/N$ are presented under *normal* drift tube conditions. Figure 2 provides a graphical representation of these product ion distributions in terms of counts per second (cps) and branching percentages for the reactions of (a) $H_3O^+$, (b) $O_2^{+\bullet}$ and (c) $NO^+$ with propofol ($C_{12}H_{18}O$). Note that no allowance in the values given has been made for any $m/z$ transmission or detection sensitivity dependencies.

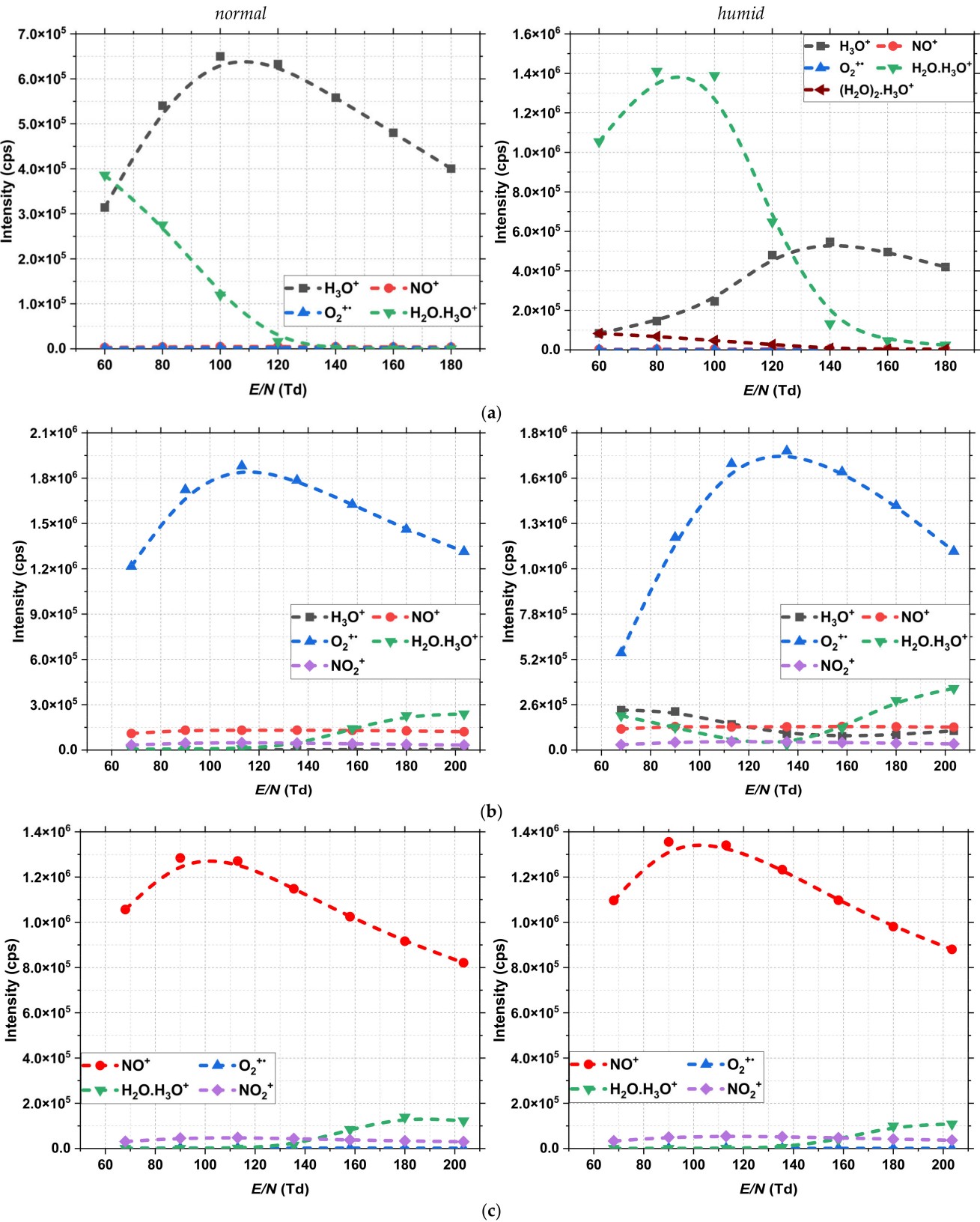

**Figure 1.** The reagent ion intensities in counts per second as a function of the reduced electric field over a range of approximately 60–210 Td for (**a**) $H_3O^+$ and $H_3O^+.H_2O$, (**b**) $O_2^{+\bullet}$ and (**c**) $NO^+$ under *normal* and *humid* drift tube operating conditions.

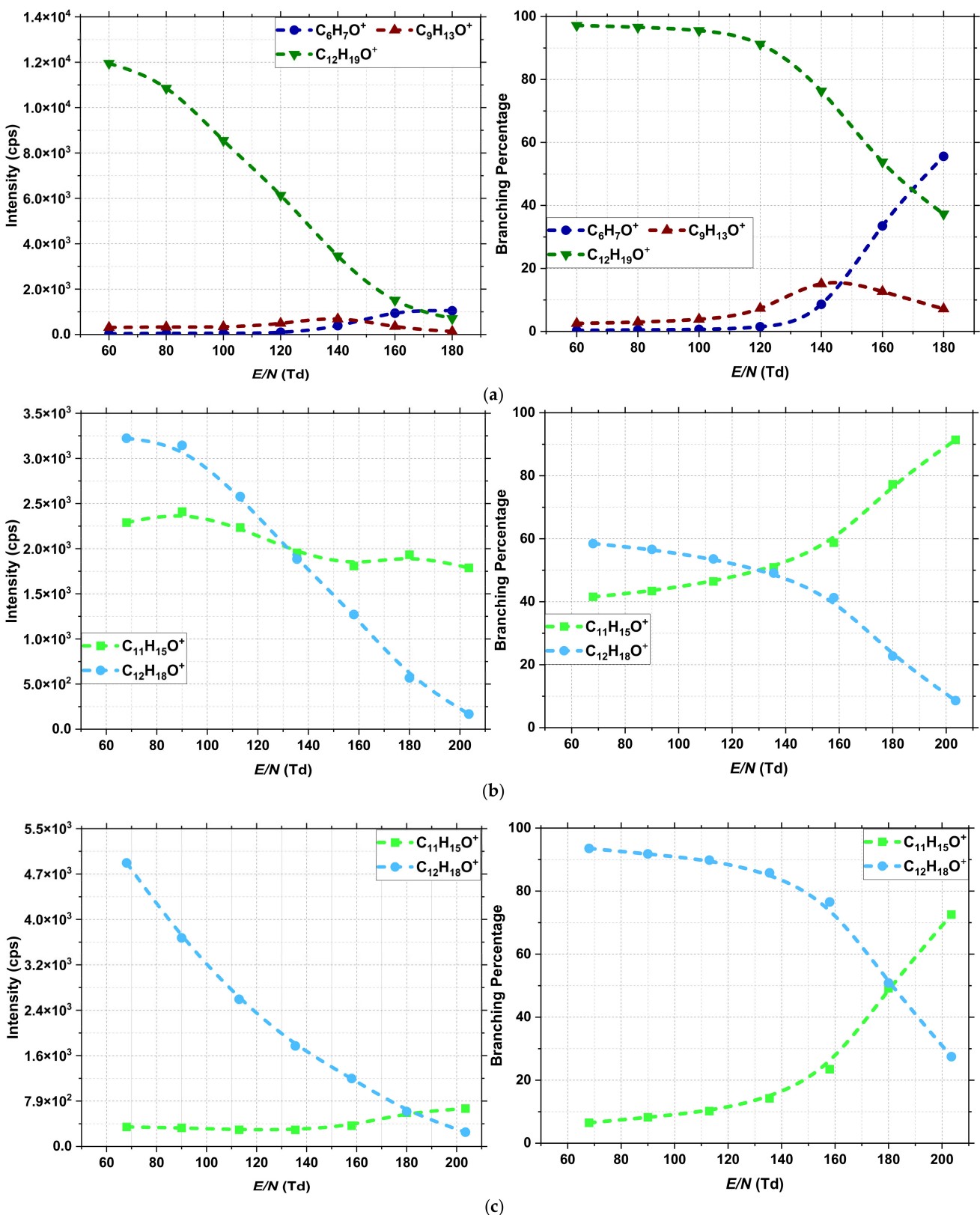

**Figure 2.** The product ion intensities given in counts per second (cps) and product ion branching percentages resulting from the reactions of (**a**) $H_3O^+$, (**b**) $O_2^{+\bullet}$ and (**c**) $NO^+$ with propofol operating the drift under *normal* humidity conditions as a function of the reduced electric field over a range of approximately 60–210 Td.

### 3.1. $H_3O^+$ Reagent Ion Measurements with Propofol

Given that the proton affinity (PA) and gas phase basicity (GB) for propofol are not available in the literature, we have used the Gaussian09W program with the GaussView05 for Windows interface and the B3LYP functional with 6–31 + G (d,p) basis set at 298 K [31] to provide calculated values. We have calculated that the PA and GB of propofol at 298 K are 770 kJ mol$^{-1}$ and 747 kJ mol$^{-1}$, respectively, which are higher than the corresponding calculated values for the water monomer (PA 684 kJ mol$^{-1}$ and GB 653 kJ mol$^{-1}$), but less than those values for the water dimer (PA 842 kJ mol$^{-1}$ and GB 777 kJ mol$^{-1}$). Thus, of the two major reagent ions present in the drift tube of the PTR-ToF-MS, namely $H_3O^+$ and $H_3O^+.H_2O$ (see Figure 1a), only $H_3O^+$ can efficiently transfer a proton to propofol. This proton transfer results in the protonated parent, $C_{12}H_{19}O^+$, at *m/z* 179.144, which is found to dominate the product ion distribution up to about 170 Td. Above 170 Td protonated phenol, $C_6H_7O^+$ at *m/z* 95.050, resulting from collision-induced dissociation processes, becomes the dominant product ion. One other product ion, also resulting from collision-induced dissociation, is observed at *m/z* 137.097, which is assigned to be protonated 2-(1-methylethyl)-phenol ($C_9H_{13}O^+$). Protonated 2-(1-methylethyl)-phenol results from a loss of $C_3H_6$ from protonated propofol, and then a sequential loss of another $C_3H_6$ from $C_9H_{13}O^+$ leads to the protonated phenol product. Harrison et al. [1] also report three product ions but only at nominal *m/z* values of *m/z* 179, *m/z* 137 and *m/z* 95 in their 2003 PTR-MS study. With the higher resolving power of the mass spectrometer available in this study, their molecular assignments are now confirmed.

A potential reaction pathway that should perhaps be expected following the transfer of a proton from $H_3O^+$ to propofol is that which leads to an elimination of $H_2O$, as depicted in Equation (1):

$$H_3O^+ + C_{12}H_{18}O \xrightarrow{\text{yields}} C_{12}H_{17}^+ + 2H_2O \quad \Delta H = +46 \text{ kJ mol}^{-1} \quad \Delta G = -11 \text{ kJ mol}^{-1}, \quad (1)$$

with the changes in enthalpy (ΔH) and free energy (ΔG) having been calculated using density functional theory calculations using a temperature of 298 K [31]. Although the temperature in the drift tube was maintained at a higher temperature than 298 K, namely 353 K, and the effective translational temperature of the ions is even greater due to the energy they gain in the electric field, the actual temperature used in the calculations is not crucial to the interpretation of the results. This is because the energetics of thermodynamics (and particularly ΔG rather than ΔH) are only being used as a guide to aid in the interpretation of the experimental observations. Given that the reaction pathway (1) is slightly exoergic at 298 K (ΔG = −11 kJ mol$^{-1}$), it is surprising that dissociative proton transfer leading to the product ion $C_{12}H_{17}^+$ is not observed, especially considering that the effective translational temperature of the $H_3O^+$ ions in the drift tube is substantially higher than that of 298 K. We have no explanation as to why the dissociative pathway (1) is not observed, and hence we propose to undertake further theoretical and experimental investigations in the hope to be able to shed some light on this observation.

### 3.2. $O_2^{+\bullet}$ Reagent Ion Measurements with Propofol

$O_2^{+\bullet}$ reacts with propofol via charge transfer leading to two product ions. The dominant one, up to about an *E/N* of 135 Td, is found at *m/z* 178.141, which is $C_{12}H_{18}O^{+\bullet}$ that results from non-dissociative charge transfer. Given the high recombination energy of $O_2^{+\bullet}$ (12.07 eV) [32] and the low ionisation potential of propofol (which must be less than 9.3 eV (see below)), there will be sufficient energy in the charge transfer process to result in direct dissociation (dissociative charge transfer [33]). This leads to the second observed product ion, $C_{11}H_{15}O^+$, at *m/z* 163.117, resulting from the loss of one of the methyl groups. This ion was also observed in a study of the reactions of $Hg^+$ (recombination energy 10.44 eV) with propofol [5]. In our SRI-ToF-MS, collisions of $C_{12}H_{18}O^+$ with the buffer gas molecules in the drift tube enhance $C_{11}H_{15}O^+$ production. This collision-induced fragmentation enhancement results in the fragment ion becoming the dominant ionic species above about 135 Td.

Charge transfer processes often lead to product ions similar to those that are dominant in electron ionisation (EI) mass spectrometry (MS). This is the case for propofol, with our results being in good agreement with the 70 eV EI-MS spectrum of propofol [34]. Although the ratio of the two major ions at the nominal *m/z* values of 163 and 178 from the EI mass spectrum differ from our observations, owing to differences in the internal energies imparted, the two mass spectra indicate the facile loss of one of the methyl groups from the molecular radical parent cation.

### 3.3. NO⁺ Reagent Ion Measurements with Propofol

The same two product ions found for the reactions of $O_2^{+\bullet}$ with propofol result from $NO^+$, although the fragment ion, $C_{11}H_{15}O^+$, is significantly less pronounced and only becomes the dominant product ion at a high *E/N* value of about 180 Td. The less pronounced fragmentation compared to the $O_2^{+\bullet}$ reactions at all *E/N* values must be a result of the smaller internal energy resulting from the transfer of charge. This result is in good agreement with the SIFT-MS study reported by Boshier et al. [6], for which the reaction of $NO^+$ with propofol resulted only in non-dissociative charge transfer. The lack of a reaction of propofol with $NO^+$ via a charge transfer process provides an upper limit for its ionisation potential, namely that it is less than that of NO, which is at 9.3 eV [35].

### 3.4. H₃O⁺, H₃O⁺.H₂O, O₂⁺• and NO⁺ Reagent Ion Measurements with 2,6-Diisopropyl-1,4-benzoquinone

No reaction of $O_2^{+\bullet}$ or $NO^+$ with 2,6-diisopropyl-1,4-benzoquinone was observed. No values for the ionisation energy of this molecule are available in the literature. However, the lack of any charge transfer suggests that the ionisation energy of the metabolite is higher than that of $O_2$, which is 12.07 eV [32]. Similarly, the PA and GB of 2,6-diisopropyl-1,4-benzoquinone are unavailable. Therefore, these values have been calculated using the same method as described for propofol. In the case of 2,6-diisopropyl-1,4-benzoquinone, there are two sites for protonation (positions 1 and 4). For position 1, the proton affinity and gas-phase basicity are calculated to be 829 kJ mol$^{-1}$ and 797 kJ mol$^{-1}$, respectively. For position 4, the proton affinity and gas-phase basicity are determined to be 865 kJ mol$^{-1}$ and 834 kJ mol$^{-1}$, respectively. These values are in agreement with our observations that both $H_3O^+$ (calculated PA($H_2O$) 684 kJ mol$^{-1}$ and GB($H_2O$) 653 kJ mol$^{-1}$) and $H_3O^+.H_2O$ (calculated PA(($H_2O)_2$) 842 kJ mol$^{-1}$ and GB(($H_2O)_2$) 777 kJ mol$^{-1}$)) reagent ions react efficiently with the metabolite. Only one product ion at *m/z* 193.124 was observed at any *E/N* value, corresponding to the protonated metabolite.

## 4. Concluding Remarks

This study provides fundamental information on the ion-molecule chemistry associated with the development of soft chemical ionisation mass spectrometric analytical techniques for the real-time monitoring of propofol and its major metabolite in exhaled breath. This is of potential benefit to developing user-friendly clinical analytical instrumentation for use in surgery and for examining metabolic processes. Detailed knowledge of the ion-molecule chemistry occurring under different operational conditions is crucial for the development of such instrumentation.

In this paper, we have presented results from the first detailed PTR/SRI-ToF-MS investigations of propofol and 2,6-diisopropyl-1,4-benzoquinone. The analyses of the reactions of the reagent ions $H_3O^+$ and $H_3O^+.H_2O$ with propofol and 2,6-diisopropyl-1,4-benzoquinone have been aided by density function calculations used to determine the proton affinities and gas-phase basicities of both compounds.

A key outcome of this study is that we have demonstrated that soft chemical ionisation mass spectrometric techniques can be used to detect propofol and its major metabolite with a high level of confidence and in real-time through the monitoring of specific distinct product ions. However, to measure propofol and 2,6-diisopropyl-1,4-benzoquinone simultaneously on breath, only $H_3O^+$ can be used as the reagent ion, the reaction of which leads

to the protonated parent species, with the maximum analytical sensitivity for these product ions being found at the lowest $E/N$ value investigated, namely 60 Td.

**Supplementary Materials:** The following supporting information can be downloaded at: https://www.mdpi.com/article/10.3390/app13074623/s1, Figure S1: NMR spectra of 2,6-diisopropyl-1,4-benzoquinone, $^1$H-NMR (400 MHz, DMSO-$d_6$, 25 °C): δ 6.54 (singlet, 2H, 2× CH (arom.)); 2.97 (septet, 2H, 2× CH (iPr); 1.08 (duplet, 12H, 2× CH$_3$ (iPr).

**Author Contributions:** A.C. developed the experimental methods, undertook all measurements, analysed the data and was involved in writing all drafts of the paper. F.W. assisted with the experimental measurements and in the interpretation of the data. S.J. assisted with the PTR-ToF-MS measurements and the analysis of the data. A.M. aided A.C. in synthesising 2,6-diisopropyl-1,4-benzoquinone and the NMR analysis for determining its purity. C.A.M. proposed the PTR-ToF-MS/SRI-ToF-MS propofol and 2,6-diisopropyl-1,4-benzoquinone study. C.A.M. also supervised A.C. and F.W. and undertook the lead in writing all draft versions of the paper. All authors have read and agreed to the published version of the manuscript.

**Funding:** We wish to acknowledge the FFG IKT der Zukunft, IKT der Zukunft, IKT der Zukunft-Resilienz und Distancing for the support of this study through the "Screening of Infection by Chemical Evaluation of Breath Volatiles for Rapid Deployment during Viral Outbreaks" DEVICE for the funding of F.W.'s Ph.D. programme and the EU HORIZON Innovation Actions HORI-ZON-CL3-2021-DRS-01-05, Project Number 101073924 (ONELAB) for funding A.C.'s Ph.D. programme.

**Institutional Review Board Statement:** Not applicable.

**Informed Consent Statement:** Not applicable.

**Data Availability Statement:** Please contact the corresponding author Anesu Chawaguta.

**Acknowledgments:** We wish to thank Peter Watts (Institute for Breath Research, University of Innsbruck, Austria) for a number of useful discussions and especially for undertaking the DFT calculations presented in this paper.

**Conflicts of Interest:** The authors declare no conflict of interest. The funders had no role in the design of the study; in the collection, analyses, or interpretation of data; in the writing of the manuscript; or in the decision to publish the results.

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
