# Peer review of "Selective Reagent Ion-Time-of-Flight-Mass Spectrometric Investigations of the Intravenous Anaesthetic Propofol and Its Major Metabolite 2,6-Diisopropyl-1,4-benzoquinone"

_applsci, doi:10.3390/app13074623_

Round 1

Reviewer 1 Report

This manuscript describes the results of a topical and important study.

The quality of the discussion and the figures is excellent and recommend this article for publication in Applied Sciences.

Author Response

We thank the reviewer for their kind comments.

Reviewer 2 Report

The manuscript by Mayhew and co-workers describes a fundamental study of Propofol ionization in the gas phase, using selective reagent ion as the chemical ionization methodology.

Propofol is widely used in surgery procedures and the development of real-time measurement devices/methodologies would represent an important advance for the safe use of anesthetic drugs. One of the most promising approaches to the real-time determination of propofol during surgery is its determination in the patient’s breath. By studying the fundamentals of gas phase chemistry and ionization of propofol, the current manuscript gives fundamental steps for turning this into a real possibility.

The study builds on the seminal work by Harrison et al., but presents new data and by extending the electric field range of values allows for a greater comprehension of the chemical ionization process of propofol. Overall, the study is well designed and the conclusions supported by the presented results. Being so, my advice is for the manuscript to be published in its current form. I have however, a minor suggestion for the authors:

-         Please do not use the word “determined” for values which have been obtained from computer based simulations with GaussView, as this may lead the less careful reader into mistakes. For instance, in lane 214 page 5: “We have determined the PA and GB of propofol…” I think it would be more correct to say “estimate”, “calculate” or similar phrasing.

Author Response

We thank the reviewer for their useful comments. We have changed determnined to calculated.

Reviewer 3 Report

This manuscript examines 2,6-diisopropylphenol (propofol, used in anaesthesi) and its metabolite 2,6-diisopropyl-1,4-benzoquinone, using specialized mass spectrometry techniques complemented by DFT computations.

It is a careful, detailed study and I think that the manuscript is appropriate for publication in Applied Sciences, within Special Issue "Application of Gas Phase Ion Chemistry", after revision of the following:

Page 5 - Concerning non-observation of dissociative pathway (1), could it just be that proton transfer, being clearly more exoergonic (deltaG -94 kJ/mol), simply dominates? A comment could be added to the discussion.

Page 6 - From non-observation of reaction of O2+• with 2,6-diisopropyl-1,4-benzoquinone, the authors suggest that its IE is higher than that of O2 (12.07 eV). This seems unwarrented if we look at the known IE of 1,4-benzoquinone which is 10 eV (https://webbook.nist.gov/chemistry/). A comment could be added to the discussion.

Author Response

We thank the reviewer for their comments. The reviewer in particular requests the following:

"Concerning non-observation of dissociative pathway (1), could it just be that proton transfer, being clearly more exoergonic (deltaG -94 kJ/mol), simply dominates? A comment could be added to the discussion."

We do not consider this is necessary, because exothermicity of the reaction does not necessarily dictate the reaction pathways in ion-molecule reactions.

The second point raised by the referee is:

"From non-observation of reaction of O2+• with 2,6-diisopropyl-1,4-benzoquinone, the authors suggest that its IE is higher than that of O2 (12.07 eV). This seems unwarrented if we look at the known IE of 1,4-benzoquinone which is 10 eV".

This is interesting, an dtherefore we would argue the opposite that our comment on the possible ionisation energy is appropriate.

We therefore kindly request that we keep the script as is.